# Aligning consciousness science and U.S. funding agency priorities

Nora A. Bradford, Angela Shen, Brian Odegaard & Megan A. K. Peters

We recently completed the Fund Consciousness Science! Project: a workshop and subawards program aimed to align United States federal funding mechanisms and consciousness research. Here we describe the project's motivation, execution, and outcomes to motivate similar efforts both locally and globally.

Consciousness science seeks to reveal and explain the neural and computational processes that give rise to consciousness (phenomenological experience). Unfortunately, though, the field faces ongoing challenges to its perceived validity in the global science community[1,2], in part due to its target of study: our intrinsic, first-person subjective experiences[3], which appear to defy empirical scientific study by definition[4]. Given its ambitious aims and ongoing struggle for recognition by scientists, consciousness science may also, at first glance, seem out of the realm of mainstream funding interests. This creates a vicious cycle, in which the relative lack of funding allocation (especially in the United States, as discussed below) contributes to challenges with the perceived legitimacy of the field. However, consciousness science has the potential to further many funding agencies' goals – and, consequently, realize its potential – if given the opportunity[5].

The study of phenomenology and consciousness clearly bears on important, topical conversations from basic through applied science. For example, consciousness researchers may be interested in experiences of sensory stimuli and the brain's own ongoing state fluctuations, as well as the difference between wakefulness, minimal consciousness, and vegetative state. Findings from consciousness science also have moral and ethical implications for policy decisions about topics like animal welfare and medical practice. For example, a patient phenomenologically feeling pain despite showing no "medical source" of pain is a critical component of research on fibromyalgia and the stigma surrounding the disease[6,7]. Likewise, pharmacological interventions for anxiety disorders which may reduce behavioral signatures of fear in animal tests can fail to generalize to human populations because they do not remedy the feeling of anxiety in people[8–10]. In our view, there is thus clear alignment between the goals of consciousness science and the priorities of major funding agencies, which include investigating decision-making under uncertainty, understanding brain connectivity in patient populations, developing brain-inspired machine learning tools, and increasing the effectiveness of cognitive training paradigms.

Unfortunately, however, the current state of funding for consciousness research in the United States is challenging[2,5], with significant disparities in the rates of public/government (as opposed to private foundation) funding for consciousness-related research in the United States compared to other geographical regions. For example (Fig. 1a), ~32% of studies in the ConTraSt database[11], which catalogs information about recent studies that purport to support or challenge current theories of consciousness, include U.S.-based sites, while ~75% include European sites including the United Kingdom (many studies include several sites). However, only ~10% of studies claim U.S. sites without international collaborators. This means that, due to the relatively few U.S.-only sites, just over 7% of the studies in the database represent U.S.-only government funded projects (i.e., p(government funding | U.S.-only)). In contrast, this number is over 34% for Europe-only studies (i.e., p(government funding | Europe-only)). Only 12.15% of the studies in the database include sites in Asia and less than half of those represent Asia-only sites. While very few studies from the database represented only Asian institutions, about 83% of those were government funded. ~26% of studies in the database do not include clear funding information. We believe this perceived misalignment between stated government funding priorities and consciousness research done in the U.S. is in error, though, so remedying it may help alleviate this funding disparity and in turn allow for consciousness science to even more meaningfully contribute scientific insight and practical benefit.

We believe a critical factor in US-based consciousness researchers' difficulty in securing mainstream funding is a lack of alignment between the way consciousness researchers talk about their research, and the way agencies' funding priorities are stated and understood by review panels. Remedying this problem could thus substantively contribute to alleviating larger challenges with perceived legitimacy of the field[1,2,5]. Further, while many of the takeaways from this project are unique to US-based researchers given its original target audience, we hope that global researchers in consciousness science as well as other fields will also benefit from them.

Crucially, this alignment problem is likely not unique to consciousness science. So, here we use consciousness science as a case study in hopes that our approach and findings will also be useful to other fields.

## The Fund Consciousness Science! Project

To remedy this situation, we ran the Fund Consciousness Science! Project[12]. The project was funded by the Templeton World Charity Foundation and took the form of a two-day workshop plus subawards to provide pilot data for a later NIH R01 proposal or similar submission – all with the goal of aligning U.S.-based consciousness science with mainstream funding mechanisms.

**Workshop and project overview.** In March 2022, 20 U.S.-based early career (pre-tenure faculty and postdoctoral) consciousness scientists (attendees) convened in Washington, D.C. with seven expert panelists (principal investigators who have successfully competed for NIH and/or NSF funding) and four program officers from the National Institutes of Health (NIH), National Science Foundation (NSF), Office of Naval Research (ONR), and Air Force Office of Scientific Research (AFOSR) who oversee relevant programs.

At the workshop, we first aimed to familiarize attendees, expert panelists, and program officers with the attendees' ongoing work; the

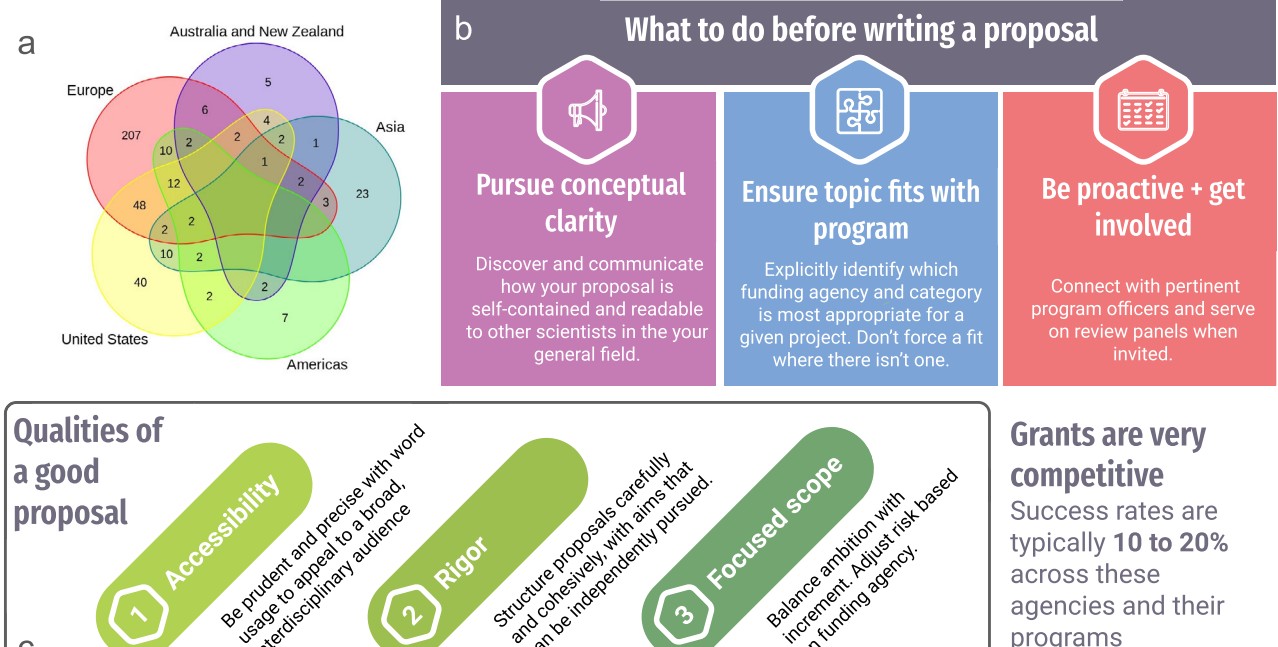

**Fig. 1 | Increasing Alignment between Consciousness Research and Funding Agencies. a** Venn diagram of locations of studies reported in the ConTraSt database[11]. 'Europe' consists of all European countries mentioned in the database, including the United Kingdom; 'Asia' includes Japan, Taiwan, China, and Korea. 'Americas' includes countries in North, Central, and South America excluding the United States. 'Australia & New Zealand' is self-explanatory. **b** Key lessons from workshop for pre-proposal writing stage. **c** Most important qualities of a proposal summarized from expert panelist and program officer presentations.

two-day workshop thus began with a flash talk by each attendee about their research. This was followed by a panel discussion with expert panelists, who shared their experiences seeking NIH and NSF funding, and presentations from program officers about their respective programs, study sections, and successful proposal practices.

We next devoted much time to breakout sessions, in which attendees worked in small groups to develop a project pitch. During these breakouts, groups also received ongoing feedback from program officers and expert panelists as they refined their projects. Groups were then given ten minutes to pitch projects with alignment goals in mind, and received feedback from program officers, expert panelists, and other attendees. Feedback included discussion of agency priorities as understood by the groups versus the program officers, questions about interdependence of aims, and clarification of specialized vocabulary. A more detailed workshop schedule is available on the project website[12].

After the workshop, attendees worked to refine their ideas and project pitches. Some groups chose to split or re-form, and all groups (original or new) were invited to submit proposals (taking the form of an NIH R21 or similar) for seed funding to produce pilot data inspired by the workshop's discussions. Proposals were assessed by a review panel consisting of expert panelists, one external reviewer, and the project PIs (Peters and Odegaard). Based on these reviews, eight projects were selected for seed funding, totalling approximately $150,000 in subawards[12].

**Lessons on alignment and grantsmanship strategy.** Throughout the workshop, all participants strove to identify how to maximize chances of success for proposals on topics related to consciousness science. These efforts focused on identifying points of overlap between attendees' research

and current priorities of U.S.-based funding mechanisms, as well as highlighting the competitive nature of the grantmaking process (success rates are typically 10 to 20% across the represented agencies and their programs[13,14]). Although we discuss how many of these lessons may be especially important for consciousness science given its specific challenges, we believe these main themes will be of utility to readers across a wide range of disciplines in maximizing the chances of success (summarized in Fig. 1b, c). A summary of advice from program officers and expert panelists is also available in PDF format at https://osf.io/d824a/.

**What to do before writing a proposal**

**Clarify field-specific concepts.** A primary source of the seeming mismatch between the goals of consciousness science and those of funding agencies may largely be due simply to a lack of shared vocabulary. So, before writing, it is imperative that you define clearly – for yourself as well as for your intended audience – the specific terminology and concepts you will use. This is especially important for consciousness science because of the broad usage of its vocabulary across many different disciplines. For example, even the word "consciousness" itself may take on different meanings for different reviewers (e.g., referring to functionalism[13,14], wakefulness vs. coma or anesthesia[15–20], phenomenal vs. access consciousness[21–23], or sentience[24,25] (especially in the context of animal consciousness; e.g.[26–28],)) in the absence of a specific and clear definition.

**Build communication skills.** Once you are clear in your definitions, you should next take care that those concepts are clear and consistent throughout all communications regarding your grant, whether it be in the

written grant itself or in communication with your program officer (see next section). This includes avoiding field-specific jargon, such as 'supervenience', 'qualia', or 'phenomenology'. Given the low chance that the members of the review panel will be in your immediate niche (conscious science specifically), it's therefore very important to ensure the proposal is self-contained and readable to other scientists in your more general field. There are many opportunities to improve science communication skills in targeting varying audiences, including stand-alone workshops (such as ComSciCon (www.comscicon.org) for graduate students or the Op-Ed project (www.theopedproject.org) and tutorials run at conferences and within universities. Researchers can also consider consulting with a writing center or communications office on campus to improve their public communication. Grantsmanship in general is a communication skill that takes time and practice to build, so start early and practice often.

**Be proactive and get involved**. Before you begin to write in earnest, connect with pertinent program officers to ensure you can articulate how your work accords with their program's priorities. You can contact program officers by sending a "white paper", which briefly details the proposed research. White papers are unofficial, have a low barrier of entry, and, in the case of AFOSR and ONR, can be emailed to program officers at any time to be discussed. This can be a helpful way to receive feedback before submitting a formal proposal and allows the program officers to gain familiarity with the work, both of which are instrumental in aligning the work with the program's interests. Connecting with a program officer ahead of writing your proposal can also ensure that the conceptual and vocabulary choices you have made will make sense to someone outside the core of the consciousness science field. Finally, white papers also enable program officers to better argue in favor of the proposal when appropriate (e.g., to institute directors and advisory councils) and to track its progress.

Another helpful way to learn about a program's priorities is to serve on a review panel to learn how other researchers target and interact with those priorities in their own proposals. Specifically, sitting on a grant panel will allow you to better understand how nuanced concepts within consciousness science can be shown to align with both agency priorities and panel discussion norms. Your participation will also greatly be appreciated by your program officer, since the fraction of people who say yes when invited to serve on a grant review panel is actually quite small.

**Know how your topic and proposed execution plan fit with the program**. In addition to speaking with program officers and sitting on grant panels, you should also seek other information to ensure a good topical fit with your target program. This is particularly important for consciousness science given its high level of interdisciplinarity. The four program officers at our workshop highlighted topics that are currently of particular interest via their funding mechanisms: the intersection between neuroscience and AI, neural coding and information representation, attention and learning (and the enhancement of both), neurotechnologies, networks (neural, social, artificial), and motor action and control. It is clear from this list that consciousness research can likely fit into any of these categories, but the specific way in which it does may be inscrutable to a panelist outside the core field; it is thus crucially important to explicitly identify and communicate which category is most appropriate for a given project.

You should also ensure that you choose a funding agency that funds the type of research you're interested in pursuing. For example, NSF tends not to fund translational research and the ONR requires an applied research

component. Consciousness science can fall into any of these categories – applied, basic science, or translational – but clear specification of the project's goals will help you decide whether it is a good fit for a given program.

**Do not reuse irrelevant work**. Relatedly, while it is tempting to "dust off" failed proposals to resubmit elsewhere, your decision to do so must be weighed against the fit of the proposal with the target program. Sometimes, a proposal is rejected due to a failure of fit, but in cases where fit was appropriate but other issues prevented the proposal from being funded, simply shipping the grant to a different agency as-is is unlikely to result in success – especially in consciousness science, where differing definitions may cause extra confusion, as described above. Re-use of ideas and projects can be a viable and efficient strategy of course, but must be done with keen attention to alignment with the new target program's scope and priorities as well as careful and specific vocabulary usage.

**Do you need preliminary data?**. In some programs, preliminary data are not required, but would certainly strengthen the proposal. In other programs, preliminary data are explicitly required, or are explicitly disallowed. Find out your target program's policy on preliminary data before crafting a proposal. In addition to web-based information and grant support services at your institution, program officers can also clarify whether and what preliminary data are appropriate.

## What makes a good proposal?
**Accessibility**. As discussed above, a key part of communicating your work and its importance is making your proposal accessible to readers from potentially widely varying fields (remember: consciousness science spans philosophy, medicine, animal biology, and more). So, in the actual writing of your proposal, be prudent and precise with word usage to appeal to a broad, interdisciplinary audience, because the expertise required to render critical evaluation of a consciousness science proposal is often more variable than for a more traditional or discipline-specific proposal. We recommend precise and descriptive vocabulary such as "conscious perception" or "perceptual awareness" that is accompanied by explicit definitions and operationalizations. Relatedly, the reviewers may not already know the significance or importance of the research, so explicitly include this in your proposal.

**Rigor**. Although not specific to consciousness science, any good proposal should be carefully and cohesively structured to propose original and novel designs or analyses; pure replications of previous research are not typically funded. A proposal's specific aims or individual components should be coherent and related, but not be so interdependent as to rely on each other's outcome. Create aims that can be independently pursued – and be independently informative – even if one fails. Aims also cannot be inconclusive; you should be able to learn something regardless of their outcomes. For each experiment, clearly state the controls, possible outcomes, implications (including for alternative theories), and any remaining unanswerable questions. Note that scientific rigor is also recently becoming a significant priority for agencies like the National Institutes of Health, which funds initiatives such as the Community for Rigor (www.c4r.io); this means that it is especially important that consciousness science proposals demonstrate exemplary rigor as we strive to establish legitimacy in the eyes of mainstream scientific funding agencies.

**Scope**. All granting agencies want to fund high-impact work, although risk tolerance varies between agencies and within programs. Depending on their respective budget sizes, some agencies want to fund big ideas,

while others prefer more incremental research. Regardless of the agency of application, it is important to write a focused proposal that balances ambition with increment, including attending to the feasibility of the project given the personnel and timeline you propose. An important step in demonstrating that your project's scope and feasibility are appropriate is continued attention to clarifying your concepts and vocabulary for a varied audience, as described above. Make sure to also highlight your own relevant training and credentials – for consciousness researchers, this may include not just stating but actually celebrating your inter-disciplinary expertise – and include any co-investigators whose expertise covers any gaps in your knowledge base. It is important that you do not give the impression that you have to learn an entire new field in order for the grant to be successful. For some agencies and programs, it is also important to write explicitly about the broader impact of the proposed research beyond the scientific, which can include societal contributions, outreach, public dissemination through workshops and the like, training and mentoring, and diversity and infrastructure. Finally, the proposed budget and timeline should be commensurate with size and scope of project.

**Summary of lessons.** The panelists and program officers offered a wide variety of helpful tips for both preparing and writing high quality grant proposals. As summarized in Fig. 1b, the panelists suggested three main ways to get prepared before writing a grant proposal: (1) pursue conceptual clarity in your thinking and communication; (2) be proactive and get involved, including sitting on a review panel and/or reaching out to program officers; (3) discover and communicate how your research topic fits well with the program of interest. The panelists and program officers also suggested that, once the writing begins, scientists should focus on three main qualities of their proposal: accessibility, rigor, and a focused scope (Fig. 1c). If a proposal is strong in these three categories, it will have a higher chance of success.

## Targeted workshops for the Association for the Scientific Study of Consciousness community

Due to the positive feedback, we decided to distill lessons from this program into a three-hour tutorial, which would allow us to share these insights with the broader consciousness science community and especially with an earlier set of trainees (students and earlier postdoctoral researchers). Thus, at the 2023 and 2024 Association for the Scientific Study of Consciousness annual meetings, we led "Communicating consciousness science to funders and the general public" tutorials to provide participants with guidance for applying for funding and led interactive science communication activities. Through these tutorials, we aimed to encourage early career researchers to share their work with a wide audience and successfully align their research proposals with the goals of funding agencies. We hope that the insights we have shared in this piece can serve as a guide to others who wish to run similar workshops for their own fields in the future. To facilitate this goal, we also share a condensed version of the slides used in those workshops in PDF format at https://osf.io/d824a/.

## Conclusion & call to action

We hope this guidance is useful for not only consciousness researchers, but investigators from all areas of science. In an age where funding pressures may contribute to hyperbole and failing to accurately expressing knowns and unknowns in specific fields, our hope is that the Fund Consciousness Science! Project and this piece can help scientists identify synergies between specialized fields and funders, to open opportunities to under-funded areas of inquiry.

## Data availability

A version of the ConTraSt database[11] annotated to support the analyses presented in this paper can be found at https://osf.io/d824a/.

**Nora A. Bradford** [1,5] ✉**, Angela Shen**[1,5], **Brian Odegaard**[2,6] **& Megan A. K. Peters** [1,3,4,6] ✉

¹Department of Cognitive Sciences, University of California Irvine, Irvine, CA, USA. ²Department of Psychology, University of Florida, Gainesville, FL, USA. ³Center for the Neurobiology of Learning and Memory, University of California Irvine, Irvine, CA, USA. ⁴Program in Brain, Mind, & Consciousness, Canadian Institute for Advanced Research, Toronto, ON, Canada. ⁵These authors contributed equally: Nora A. Bradford, Angela Shen. ⁶These authors jointly supervised this work: Brian Odegaard, Megan A. K. Peters. ✉e-mail: noraab@uci.edu; megan.peters@uci.edu

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

## Acknowledgements
We thank Tomàs Ortega for assistance with graphics. Funding for the Fund Consciousness Science! project was provided by the Templeton World Charity Foundation (TWCF 0495). MAKP's effort was also supported as a Fellow in the Canadian Institute for Advanced Research (CIFAR) Brain Mind & Consciousness program. While TWCF funded the project and its subawards in the original proposed form, neither TWCF nor CIFAR played a role in planning and execution of the workshop nor selection of attendees' project proposals for seed funding awards.

## Author contributions
MAKP and BO proposed, planned, and ran the workshop. NB and AS assisted with workshop coordination. NB, AS, BO, and MAKP wrote the manuscript. NB created the figure.

## Competing interests
The authors declare no competing interest.
