## [Transparent Peer Review file · Communications Biology]

Aligning consciousness science and U.S. funding agency priorities

Corresponding Author: Mx Nora Bradford

Version 0:

Reviewer comments:

Reviewer #1

(Remarks to the Author)

The manuscript by Bradford and colleagues is a commentary meant to provide a summary of a workshop about a workshop meant to foster the development of funding for consciousness science in the US. The manuscript is divided in 3 main sections: an introductory section on why funding consciousness science is relevant, a recap of the workshop and a summary of recommendations for improving grant applications in the field.

I found the commentary moderately relevant, as it ranges between being very focused to excessively broad. As a researcher working in the field of consciousness, I was left to wonder what I learned from the commentary, and I believe it would be most appropriate for a regional publication focusing on the US funding system. Below my main recommendations:

MAJOR:

1) The introduction focuses on a discussion of the US and European funding for consciousness science. In view of the global reach of the journal, it would have been important, in my opinion, to at least mention the funding situation in other areas of the world. This would have made the manuscript more relevant outside the US, also because the authors then simply focus on the US system. Of course the reason is the specific nature of the project, but this is in my opinion a reason to focus on a regional publication venue.

2) I did not fully understand why the authors described in depth the dynamics of the workshop (including the names of all participants), but only put in an external link the much more relevant recommendations from program officers and panelists. I found the latter to be an useful resource for researchers aiming to find a funding instrument. I think that these recommendations should have been an integral part (if not the focus) of the manuscript.

3) The sections on lessons of alignment and grantmanship is (in my opinion) very broad and not specific focused on consciousness. In fact, it is applicable to any field and contains interesting but rather basic recommendations. I think it is an useful resource for any young investigators moving their first steps in the US funding systems (as most lessons specifically apply to the US system), but I failed to grasp the specific and unique relevant to the field of consciousness science.

MINOR:

- I found a typo, where the word principle was used instead of principal (investigator)

Reviewer #2

(Remarks to the Author)

This is a good and useful article describing the Fund Consciousness Science project. Some of the data is good to have and informative (notably, data about the funding differences between Europe and the US). Describing what happened at the workshop is also useful and will increase the impact of this important project, as others might try to replicate it (perhaps in other fields).

I just have two very minor comments:

(1) Could the authors just provide an example of 'misalignment of vocabulary'? The discussion seemed a bit abstract and a

concrete example could help.

(2) The advice to 'build communication skills' is not very useful without providing advice on how to do that.

Version 1:

Reviewer comments:

Reviewer #1

(Remarks to the Author)

The authors partially addressed my concerns. In particular, the manuscript recommendations remain broadly applicable to any scientific field rather than provide specific guidance to consciousness scientists. Below I outline my opinion on the authors responses:

1) International relevance of the manuscript.

The authors have added a note about the status of consciousness funding outside the US and Europe. However, the rest of the recommendations has been left unchanged and still refers to the US system alone. Although I find the authors' response sufficient, I believe that the focus on the US system should be made more evident by e.g. changing the title of the manuscript to reflect this fact.

2) Recommendation and workshop dynamics.

While the authors added a summary of the workshop recommendation, my main comments were not addressed. Specifically, I had asked the authors to explain "why the authors described in depth the dynamics of the workshop (including the names of all participants)" and why they instead left out the more detailed (and in my opinion more useful) information presented in the linked documents

3) Specificity to the consciousness field

The authors now modified their manuscript to specify why some issues are particularly relevant to the consciousness field. However, I believe that this does not fully answer my comment about the lack to specificity of the recommendations to the consciousness field. In fact, I completely agree with many of the authors' statements such as "Paying attention to these markers of rigor is important in any field, but is especially important in consciousness science as we strive to establish legitimacy in the eyes of mainstream scientific funding agencies". However, I think this should have been the starting point of each of their recommendations, and not just an afterthought. In other words, I think that the authors need to transform the broadly applicable recommendations to specific guidelines for the consciousness field. This requires a more careful re-writing of the recommendations compared to what the authors did in their current revision.

Reviewer #2

(Remarks to the Author)

I thank the authors for responding to my comments. I believe that the manuscript is ready now.

Version 2:

Reviewer comments:

Reviewer #1

(Remarks to the Author)

The authors satisfactorily addressed all my comments.

Response to Reviewers

Reviewer #1 (Remarks to the Author):

The manuscript by Bradford and colleagues is a commentary meant to provide a summary of a workshop about a workshop meant to foster the development of funding for consciousness science in the US. The manuscript is divided in 3 main sections: an introductory section on why funding consciousness science is relevant, a recap of the workshop and a summary of recommendations for improving grant applications in the field.

I found the commentary moderately relevant, as it ranges between being very focused to excessively broad. As a researcher working in the field of consciousness, I was left to wonder what I learned from the commentary, and I believe it would be most appropriate for a regional publication focusing on the US funding system. Below my main recommendations:

Response: We thank the Reviewer for their summary and have endeavored to make the piece more relevant to the field of consciousness research through our specific changes, described point-by-point below.

MAJOR:

1) The introduction focuses on a discussion of the US and European funding for consciousness science. In view of the global reach of the journal, it would have been important, in my opinion, to at least mention the funding situation in other areas of the world. This would have made the manuscript more relevant outside the US, also because the authors then simply focus on the US system. Of course the reason is the specific nature of the project, but this is in my opinion a reason to focus on a regional publication venue.

Response: Thank you for this point. We have added more information about consciousness science funding in Asia, as requested by the editor. The newly added sections read:

Only 12.15% of the studies in the database include sites in Asia and less than half of those represent Asia-only sites. While very few studies from the database represented only Asian institutions, about 83% of those were government funded.

2) I did not fully understand why the authors described in depth the dynamics of the workshop (including the names of all participants), but only put in an external link the much more relevant recommendations from program officers and panelists. I found the latter to be a useful resource

for researchers aiming to find a funding instrument. I think that these recommendations should have been an integral part (if not the focus) of the manuscript.

Response: We appreciate the Reviewer's concerns, but do note that in addition to the external link, we discuss in depth the recommendations from both the program officers and expert panelists in the section entitled Lessons on alignment and grantsmanship strategy. However, since this was unclear, we have now amended the text to make more clear that these sections provide direct summary and detailed information from these resources, and that the linked materials are meant to supplement the lessons described in the article by providing the primary resources shared by the program officers. The new text reads:

Summary of lessons:

The panelists and program officers offered a wide variety of helpful tips for both preparing and writing high quality grant proposals. As summarized in Figure 1b, the panelists suggested three main ways to get prepared before writing a grant proposal: (1) be proactive and get involved on a review panel, (2) ensure your research topic fits well with the program of interest, and (3) build communication skills in order to effectively relay your scientific goals to future review committees. The panelists and program officers also suggested that, once the writing begins, scientists should focus on three main qualities of their proposal: accessibility, rigor, and a focused scope (Figure 1c). If a proposal is strong in these three categories, it will have a higher chance of success.

3) The sections on lessons of alignment and grantsmanship are (in my opinion) very broad and not specific focused on consciousness. In fact, it is applicable to any field and contains interesting but rather basic recommendations. I think it is an useful resource for any young investigators moving their first steps in the US funding systems (as most lessons specifically apply to the US system), but I failed to grasp the specific and unique relevant to the field of consciousness science.

Response: We appreciate the Reviewer's concerns, and note that the balance between making suggestions specific for consciousness science while also making the piece relevant to researchers outside this niche field is delicate. While it's true that many of our recommendations apply widely, they stemmed specifically from the discussions at the workshop surrounding consciousness research. In response to this concern, we have now added additional language to highlight why each point is specifically important for consciousness science. These changes are tracked in the revised manuscript throughout, so we do not reproduce them here.

MINOR:

- I found a typo, where the word principle was used instead of principal (investigator)

Response: Thank you - we have fixed that.

Reviewer #2 (Remarks to the Author):

This is a good and useful article describing the Fund Consciousness Science project. Some of the data is good to have and informative (notably, data about the funding differences between Europe and the US). Describing what happened at the workshop is also useful and will increase the impact of this important project, as others might try to replicate it (perhaps in other fields).

Response: We thank the Reviewer for their positive assessment.

I just have two very minor comments:

(1) Could the authors just provide an example of 'misalignment of vocabulary'? The discussion seemed a bit abstract and a concrete example could help.

Response: This is a very good suggestion, thank you. We have now added an example to clarify what we mean by this misalignment. That example reads:

For example, even the word “consciousness” may take on different meanings for different reviewers (e.g., referring to functionalism^{13,14}, wakefulness vs. coma or anesthesia^{15–20}, phenomenal vs. access consciousness^{21–23}, or sentience^{24,25} (especially in the context of animal consciousness; e.g., ²⁶)) in the absence of a specific and clear definition.

(2) The advice to 'build communication skills' is not very useful without providing advice on how to do that.

Response: We appreciate the opportunity to provide specific recommendations for this suggestion. We have now added the following:

There are many opportunities to improve science communication skills, including stand-alone workshops (such as ComSciCon (www.comscicon.org) for graduate students or the Op-Ed project (www.theopedproject.org) and tutorials run at conferences and within universities. Researchers can also consider consulting with a writing center or communications office on campus to improve their public communication.

Response to Reviewers

Dear Editor,

Thank you very much for the opportunity to revise our manuscript in response to the reviewers' comments. We have addressed the suggestions from both reviewers, as well as the comments directly from the editorial team:

1) You should revise your article title to better outline the main focus on the U.S. funding system. I would suggest the following title: **"Aligning consciousness science and U.S. funding agency priorities"**.

Response: We agree and have updated the title accordingly.

2) We tend to agree that the list of the attendees should be removed from the article and transferred to the linked documents, to improve the readability.

Response: Agreed; we have removed this list.

3) Thank you for providing a more specific set of recommendations relevant to the consciousness field. However, we think that some recommendations remain very general (for instance those in Figure 1) and you should try to address this issue by providing additional specific advice. We also ask you to better emphasize what really distinguishes the consciousness field from other neuroscience research areas in regard to funding opportunities and rates.

Response: We agree and have substantially revised this section to more clearly lay out the specific components of these pieces of advice that link with consciousness science per se. All changes are tracked in the revised manuscript, and we provide more detailed comments in the point-by-point Response to Reviewers document.

In the attached, we include a point-by-point response to each Reviewer's comments, as well as a manuscript with all changes tracked. We believe the changes have significantly improved the manuscript's clarity and are grateful for the opportunity to make these changes.

Thank you again and we look forward to your assessment.

Best regards,

Nora Bradford, Angela Shen, Brian Odegaard, and Megan Peters

Reviewer #1 (Remarks to the Author):

1) International relevance of the manuscript.

The authors have added a note about the status of consciousness funding outside the US and Europe. However, the rest of the recommendations has been left unchanged and still refers to the US system alone. Although I find the authors' response sufficient, I believe that the focus on the US system should be made more evident by e.g. changing the title of the manuscript to reflect this fact.

Response: We thank the Reviewer for their suggestion. We have changed the title to “Aligning consciousness science and U.S. funding agency priorities”

2) Recommendation and workshop dynamics.

While the authors added a summary of the workshop recommendation, my main comments were not addressed. Specifically, I had asked the authors to explain "why the authors described in depth the dynamics of the workshop (including the names of all participants)" and why they instead left out the more detailed (and in my opinion more useful) information presented in the linked documents

Response: Thank you for this feedback. We have removed the list of participants from the manuscript. For the sake of keeping the document short, we'll keep the more detailed information in the linked documents; this is also consistent with previous reviews of this manuscript.

3) Specificity to the consciousness field

The authors now modified their manuscript to specify why some issues are particularly relevant to the consciousness field. However, I believe that this does not fully answer my comment about the lack of specificity of the recommendations to the consciousness field. In fact, I completely agree with many of the authors' statements such as "Paying attention to these markers of rigor is important in any field, but is especially important in consciousness science as we strive to establish legitimacy in the eyes of mainstream scientific funding agencies". However, I think this should have been the starting point of each of their recommendations, and not just an afterthought. In other words, I think that the authors need to transform the broadly applicable recommendations to specific guidelines for the consciousness field. This requires a more careful re-writing of the recommendations compared to what the authors did in their current revision.

Response: Thank you for your feedback on this point. We have addressed this concern in multiple ways.

First, we added the short sentence below in the motivation section.

Unfortunately, though, the field faces ongoing challenges to its perceived validity in the global science community^{1,2}, in part due to its target of study: our intrinsic, first-person subjective experiences³, which appear to defy empirical scientific study by definition⁴.

Additionally, we have substantially revised the Lessons on alignment and grantsmanship strategy sections to reflect a focus and prioritization of concerns related to consciousness science. Because these changes are interspersed throughout the text we do not copy them here into the Response to Reviewers document, but all changes are tracked in the revised manuscript. These changes also include an update to the figure to reflect the newly-revised section's organization and content.